# Varied Expression of Senescence-Associated and Ethylene-Related Genes during Postharvest Storage of *Brassica* Vegetables

**DOI:** 10.3390/ijms22020839

**Published:** 2021-01-15

**Authors:** Yogesh Ahlawat, Tie Liu

**Affiliations:** Department of Horticultural Sciences, University of Florida, Gainesville, FL 32611, USA

**Keywords:** ethylene, senescence, *Brassica* vegetables, postharvest technology

## Abstract

The genus *Brassica* comprises a highly diverse range of vegetable crops varying in morphology, harvestable crop product, and postharvest shelf-life that has arisen through domestication, artificial selection and plant breeding. Previous postharvest studies on the shelf-life of *Brassica* species has mainly focused on the variable rates of physiological changes including respiration and transpiration. Therefore, further understanding of the molecular basis of postharvest senescence in *Brassica* vegetables is needed to understand its progression in improving their postharvest shelf-life. The aim of this study was to better understand the trajectory of molecular responses in senescence-associated genes but not induced by ethylene and ethylene-induced genes towards altered postharvest storage conditions. After storage at different temperatures, the expression levels of the key senescence-associated genes (SAGs) and the ethylene biosynthesis, perception, and signaling genes were quantitatively analyzed in cabbage, broccoli and kale. The expression levels of these genes were tightly linked to storage temperature and phase of senescence. Expression of *ORE15*, *SAG12,* and *NAC29* were continuously increased during the twelve days of postharvest storage at room temperature. Prolonged exposure of these three vegetables to cold temperature reduced the variation in the expression levels of *ORE15* and *SAG12*, observed as mostly decreased which resulted in limiting senescence. The transcript levels of the ethylene receptor were also decreased at lower temperature, further suggesting that decreased ethylene biosynthesis and signaling in cabbage during postharvest storage would delay the senescence mechanism. These results enhanced our understanding of the transcriptional changes in ethylene-independent SAGs and ethylene-related genes in postharvest senescence, as well as the timing and temperature sensitive molecular events associated with senescence in cabbage, broccoli and kale and this knowledge can potentially be used for the improvement of postharvest storage in *Brassica* vegetables.

## 1. Introduction

Postharvest longevity of perishable produce remains a challenge in the global fresh market supply chain [1]. Postharvest longevity is determined by the rates of ripening and senescence, which are influenced by harvest time and storage conditions. Ripening and senescence are predominantly regulated by ethylene, which produces a plethora of metabolic effects within the harvested produce, leading to physiological and developmental changes during postharvest. The traditional supply chain for most perishable produce is long and complex, with postharvest losses ranging from 12–46% after seven days in storage. Therefore, it is important to improve the sustainability of postharvest quality in order to avoid losses and to bolster crop prices. In this regard, significant amounts of research have been done to study what physiological and biochemical changes during postharvest senescence in fruits and vegetables [2].

The *Brassicaceae* family contains some of the most important vegetable crops worldwide [3]. The family is widely distributed across all continents except for antarctica and characterized by the presence of several secondary metabolites like phenolics, flavonoids including certain bioactive compounds like glucosinolates and indoles [4]. Vegetables like green head cabbage, broccoli and kale are all cultivars of the species *Brassica oleracea*. Vegetables in this family have been reported to possess many health benefits since they produce several types of antioxidants and secondary metabolites, including glucosinolates and phenylpropanoids that can enhance the immune system and reduce the risk of chronic diseases [5]. However, after harvest most of these crops turn yellow rapidly and are likely to senesce during postharvest storage and transportation. Leaf senescence is a beneficial process for maintaining plant survival [6]. However, in regards to agricultural productivity, leaf senescence negatively affects crop yield and postharvest attributes such as nutrient loss and changes in the produce color via chloroplast dismantling [7]. Therefore, in order to increase shelf life and prevent economic losses, it is crucial to understand the molecular mechanisms underlying postharvest senescence in these vegetable crops [8].

Recent studies have identified several gene families that have roles during senescence [9], including the senescence-associated genes (SAGs) and two large families of transcription factors, the *NAC* (*NAM*, *ATAF* and *CUC*) and *WRKY* (have the WRKYGQK motif at the N-terminal end) families [10]. In total, more than 20 transcription factors have been found to be transcriptionally upregulated during senescence, including *ORE1* (*ORESARA1*, an ageing regulator) [11], *NAC2*, *NAC29*, and *ORS1* [12]. The *NAC* transcription factor *ORE1* plays a key role in the regulation of leaf senescence and programmed cell death in *Arabidopsis thaliana* [13]. Overexpression of *ORE1* in *Arabidopsis* causes a more rapid progression towards senescence, whereas *ORE1* knockdown lines had a slower senescence response. *ORE15*, a PLATZ (plant AT-rich sequence and Zinc-binding protein) transcription factor, is linked to developmental and senescence processes in *Arabidopsis* through its interaction with the growth regulating factor (GRF) pathway [14]. *ore15* mutants exhibited increased leaf longevity and delayed leaf yellowing as compared to wild type. *ore15* mutants showed delayed leaf senescence in response to dark, salt or oxidative stresses [15]. Expression studies have revealed that *ORE15* transcript levels were mostly concentrated in young leaves and at lower levels in ageing leaves [15]. *SAG12* is another candidate gene associated with plant development. In lettuce, expression of the *Arabidopsis* Isopentenyl Transferase (*IPT*) gene driven by the *SAG12* promoter delayed postharvest senescence through retention of chlorophyll content in outer and inner leaves [16]. Additionally, ethylene has been observed to regulate *SAG12* levels in *Arabidopsis*. The *etr1* mutant showed delayed leaf senescence that coincided with decreased levels of *SAG12* [17]. Wild-type *Arabidopsis* plants, when exogenously subjected to ethylene were observed to be exhibiting higher levels of *SAG12* and accelerated senescence in the leaves.

During normal development, the gaseous hormone ethylene plays a very important role in tissue maturation, including senescence. Plants have internal signaling pathways that produce ethylene, perceive ethylene and trigger the downstream biological responses at the time of senescence. The release of ethylene from vegetables after harvest is responsible for affecting physiological processes like increasing the respiration rate depending upon the storage strategies which enhances other processes like ripening and senescence. Artificial management of ethylene biosynthesis, perception, and signaling in plants has proven to be the most successful way to delay senescence for prolonged postharvest storage of fruits and vegetables. However, for the species in the *Brassicaceae* family, it is not clear how the signaling cascades work after harvest.

Ethylene is perceived by a large family of receptors, such as *ETR1*, *ERS1*, and *EIN4*. Mutant studies have elucidated the genetic regulation of the ethylene signaling pathway where *ETR1* (*Ethylene Response 1*) and *EIN4* (*Ethylene Insensitive 4*) acts directly upstream of *CTR1* (*Constitutive Triple Response 1*), while *EIN3, EIN4, EIN5*, and *EIN6* act downstream of *CTR1* [10]. *ETR1* and *CTR1* are two key components of the ethylene receptor complex and act as negative regulators to the downstream targets during the presence of ethylene. In peach, the ethylene receptors showed varied transcript responses at the time of ripening [18]. Ethylene production in *Arabidopsis* removes the negative regulation of *CTR1* on the downstream *EIN2* protein. *EIN2* is the first positive regulator in the ethylene signaling cascade downstream of *CTR1*, but the *EIN2* transcript does not transcriptionally change during tomato fruit ripening [19]. An increase in *EIN2* gene expression occurs at the mature green stages in tomato [19]. In peach, an induction of a putative ortholog of *EIN2* was observed during the transition from the immature to mature green stage [20]. The ethylene-induced activation of *EIN2* stabilizes the primary downstream transcription factors *EIN3* and *EIL1* that control the expression of secondary transcription factors (ERFs) and regulate certain growth and stress responses [21]. In some of the ethylene-regulated processes, ethylene functions via programmed cell death (PCD).

In addition to internal regulation, external factors such as environmental cues, mainly drought, pathogen attack, and varied temperatures can trigger ethylene signaling that induces senescence [22]. Therefore, these external cues and endogenous developmental regulators interact with each other, resulting in physiological changes leading to senescence [22]. The signaling mechanisms specific to natural and induced senescence are unclear, since varied patterns of gene expression were observed during certain metabolic and developmental processes [23]. Low temperature is one of the most effective methods for extending the postharvest shelf life of produce, because it reduces the respiration rate, ethylene production, moisture loss and plant pathogen growth [24]. However, cold temperature sometimes affects the nutritional traits of leafy vegetables [24]. Postharvest storage conditions affect the quality and nutritional parameters in vegetables by changing the chlorophyll, phenolics and ascorbic acid contents. However, it is still unclear how ethylene production and perception are regulated and how senescence is delayed during cold storage of leafy vegetables. Recently, abscisic acid treatment of harvested cabbage was shown to subdue the senescence mechanism by reducing the ethylene content by half and by maintaining the water and chlorophyll contents [25]. Ethylene receptors like *CTR1* were also downregulated with ABA treatment, reducing the downstream signaling of molecules like *EIN3* and *EIL1*, which resulted in decreased ethylene accumulation and delayed senescence.

This study was conducted to explore the effects of postharvest storage temperature in the *Brassica* vegetables cabbage, broccoli and kale through tracking the molecular events associated with senescence and ethylene signaling. Cabbage leaves, broccoli florets and kale leaves were stored under room temperature and refrigerated conditions while physiological parameters such as moisture loss and gene expression were analyzed. This study showed how regulation of ethylene perception by controlling the ambient temperature affects the signaling cascade of downstream effectors. Our study demonstrated that harvested *Brassica* crops such as cabbage, broccoli and kale senesce at different rates when subjected to similar storage conditions. Cabbage was observed to have a longer shelf life and lower senescence rate in comparison to broccoli and kale. Ethylene signaling genes were upregulated as room temperature storage in broccoli and kale, suggesting that ethylene signaling plays a key role in senescence at room temperature. Furthermore, this study showed that lower temperature inhibits the expression of ethylene biosynthetic genes, slowing the progression towards senescence.

## 2. Results

### 2.1. Effect of Storage Temperatures on the Fresh Weight and Relative Water Content of Brassica Vegetables

Postharvest senescence was studied in cabbage, broccoli and kale stored under cold and room temperatures for 12 days. Browning in these *Brassica* vegetables was the first visible symptom of senescence, likely due to chlorophyll breakdown. Under room temperature, cabbage and kale leaves first became brittle and soft on day 6 and gradually decreased in firmness during the observation period. The major visible changes were characterized by tissue browning along the midrib and veins in cabbage and kale and started with yellowing of floret spores in broccoli eventually turning into brown, occurred on day 9 in cabbage, on day 6 in broccoli, and on day 3 in kale (Figure 1). In contrast, under refrigeration, there were no visible changes in color or firmness found in cabbage, broccoli or kale over the 12 days of storage (Figure 1). To quantify the changes in tissue firmness during the postharvest storage period, fresh weight was measured during room temperature and cool storage. At the room temperature, the fresh weight in cabbage slowly decreased during storage to ~50% on day 12; however, this only decreased by 25% when stored at 4 °C (Figure 2A–C), indicating that a lower temperature significantly slowed down the decline in fresh weight in cabbage over the first 12 days of storage. The loss in fresh weight during storage was different for broccoli and kale. Fresh weight in broccoli florets decreased by 43% on day 6 and 70% on day 12 at room temperature but only by 24% (day 6) and 40% (day 12) at 4 °C. The fresh weight in kale rapidly decreased by 57% (day 6) and 69% (day 12) at room temperature, and nearly the same amount of water loss was observed during cold storage. These results suggested that the rates of fresh weight in broccoli and kale were much faster than in cabbage when stored at room temperature. Meanwhile, the percentage change in fresh weight in kale remained same despite under similar storage conditions, suggesting that cold storage has little effect on the shelf life of kale. To better characterize the physiological consequence of cellular water deficient, we measure the relative water content (RWC) to monitor the appropriate plant water status. A gradual decrease in RWC in all three vegetable were observed over the sampling period. Contrary to observations established during the percentage change in fresh weight, we monitored that the measurement of RWC (Figure 2D–F) was 1.68% on day 9 in kale in comparison to cabbage (81%) and broccoli (79%) at room temperature. Kale has significantly lower water retention capacity at room temperature even at cold storage which further supports the evidence of its shorter shelf life.

### 2.2. The Expression Levels of Specific Senescence-Associated Genes (*SAGs*) Were Highly Induced in Postharvest Cabbage, Broccoli and Kale While Stored at Room Temperature

To explore the molecular mechanisms associated with changes during postharvest storage as well as to identify factors determining the shelf life of these *Brassica* vegetables, the expression levels of select genes were determined using Real-Time PCR. Given the fact that SAGs and ethylene-related genes are responsible for driving the shelf-life and senescence processes, we focused on those two group of genes and analyzed expression patterns in cabbage, broccoli and kale stored under cold and room temperatures. Among the well-characterized SAGs, we selected three SAGs (*ORE15*, *SAG12,* and *NAC29*) that were investigated since they have been well-studied for their roles during senescence in *Arabidopsis*, cabbage, and tomato as well as acting independently of the ethylene [25,26]. Under room temperature, *ORE15* expression in cabbage increased by 1.5-fold on day 6 and by 2.5-fold on day 12 compared to day 1. With cold storage, only a 4.4-fold increase was observed between day 6 and day 12. Under either temperature, expression of *ORE15* increased progressively during the 12-day period, but the expression level was much higher at room temperature than under cold storage. This observation confirms that *ORE15* expression increased with storage time under both temperatures, but that the increase was at a lower amplitude and delayed under cold storage (Figure 3A). Interestingly, *ORE15* gradually decreased in both broccoli and kale when stored at cold temperature (Figure 3A). This observation is conflict with the weight loss data, suggesting that the shorter shelf life in broccoli and kale may be mainly due to losses in turgor pressure (Figure 2D–F) triggered by water loss instead of tissue senescence. Moreover, *NAC29,* a key senescence-associated *NAC* transcription factors exhibited transcriptional induction by 2.2-fold on day 6 and significantly increased by 4.6-fold on day 12 in cabbage at room temperature. It is worth to noting that there was little change in *NAC* expression in cabbage when stored at lower temperature (Figure 3C). Contrastingly, *NAC29* transcript levels were significantly elevated in kale and broccoli on day 12 by 7.4-fold and 6.5-fold respectively at room temperature. The expression of kale *NAC29* was higher at lower temperature on day 6 (5.5-fold) and day 12 (6.5-fold) than that of broccoli with 2.1-fold on day 6 and day 12, respectively (Figure 3C), indicating that lower temperature has little effect on transcriptional regulation of *NAC29* expression in kale. Taken together, this experiment suggested that divergent transcript expression of SAGs in *Brassica* vegetables might lead to the differential senescence progressing through complex interactions between senescence-related factors.

The expression pattern of *SAG12* was also examined (Figure 3B). *SAG12* levels were also highly up-regulated at room temperature. In broccoli, *SAG12* transcript levels were rapidly induced by 6-fold on day 6 and 8-fold on day 12 in comparison to day 1. Similarly, in kale *SAG12* increased 9-fold on day 6 and 19-fold on day 12, implying that both *SAG12* and *ORE15* were highly induced to promote senescence when stored at room temperature. The results supported the idea that kale has a higher degree of senescence in comparison to cabbage and broccoli. It is interesting to note that cold temperature storage kept *SAG12* levels remained steady in cabbage and kale, while they continuously increased in broccoli (Figure 3B). Although variation in transcript level of *SAG12* expression were not detected in these large tissues sample, changes might occur at more tissue-specific levels to alter gene expression through a different pathway during postharvest senescence. Taken together, SAGs display variation in expression patterns in the three brassica vegetables under two different storage conditions. These observations prompted us to test if ethylene affects senescence during cold storage of cabbage, broccoli and kale.

### 2.3. Expression of Ethylene-Related Genes in Harvested Cabbage, Broccoli and Kale during Storage

One of the key enzymes from the ethylene biosynthetic pathway, *ACO* (*ACC Oxidase*), is known to be responsible for most ethylene production [26]. In cabbage, the expression of *ACO* decreased by 2-fold from day 0 to day 12 when stored at 25 °C, while it significantly increased, by 1.6-fold on day 6, when stored at 4 °C, and then remained steady on day 12 (Figure 4A). This suggested that ethylene production was inhibited. In broccoli, the *ACO* transcript levels decreased by 0.5-fold by day 12 at room temperature but was more variable and then higher when at 4 °C (Figure 4A). In kale, there was a significant, 3-fold induction in *ACO* expression by day 12 when stored at room temperature, suggesting an increased production of ethylene, while under cold temperature storage, *ACO* expression slightly decreased. It is worth mentioning that *ACO* exhibited an opposite expression pattern in kale compared to cabbage and broccoli in response to the different storage temperatures. *ACO* has been reported to be the rate-limiting factor for ethylene production [26]. Similarly, in cabbage *ACS* (*ACC Synthase*), expression did not show any significant change under two storage conditions indicated that the ethylene production rates are generally very low (Figure 4B). However, in broccoli, *ACS* transcript was increased by 2.7 fold on day 6 and 5 fold on day 12 at room temperature whereas was decreased at cold temperature, suggesting that ethylene production rates were inhibited at low temperature. Interestingly, in kale, *ACS* expression was rapidly induced in room temperature from day 6 to day 12 in comparison to day 1 (Figure 4B). the expression of *ACS* when stored at low temperature. This is consistent with the observed SAGs gene expression in kale, suggesting that ethylene production accelerated the senescence in kale and thus reduced the shelf life of kale when stored at room temperature. However, the lack of significant changes in *ACO* expression under cold conditions indicated that some additional factors, such as ABA, may be involved during postharvest senescence. Future studies on the interconnections between phytohormones and their transport during postharvest storage are essential to better understand the senescence in brassica vegetables.

Subsequently, we also examined the expression of the ethylene perception factor *ETR1* and its regulator *CTR1* in these *Brassica* vegetables. The transcript levels for the ethylene receptor *ETR1* were generally lower during storage at 4 °C than at room temperature (Figure 4C). *ETR1* expression at 25 °C significantly increased on day 12, by 17-fold in cabbage. Similarly, *CTR1* levels were increased by 4-fold on day 6 and 8-fold on day 12 at room temperature, 2.6-fold in broccoli, and 7-fold in kale (Figure 4D). Interestingly, there were no significant changes observed in broccoli either on day 6 and 12. Similar observations were made in 4 °C. This suggested there could be additional ethylene perception factors in broccoli that senses the ethylene production. Surprisingly, at 4 °C, kale showed a surge in transcript levels on day 6, by 7-fold, followed by a decrease of 3-fold on day 12. Broccoli showed increased *ETR1* levels on day 12 and *CTR1* levels were 1.6-fold and kale exhibited 4-fold change in *CTR1* when stored at lower temperature (Figure 4D). This result revealed the ethylene perception in kale played an important role in ethylene-induced senescence and might be more sensitive to ethylene production than broccoli and cabbage during postharvest storage.

Ethylene promotes ripening and controls senescence through a cascade of signaling pathways. Interaction between ethylene or an ethylene analog and *ETR* negatively regulates ripening and senescence. We quantitatively analyzed downstream components of ethylene signaling pathways by examining the expression patterns of two ethylene signaling genes *EIN2* and *EIN3*. In cabbage, *EIN2* transcript levels decreased with time when stored at room temperature but were significantly increased during storage at 4 °C, by 1.5-fold on day 12 (Figure 4E). In contrast, broccoli and kale showed significant increase in *EIN2* transcript on days 6 and 12 at room temperature and decreases under cold storage (Figure 4E). *EIN3* expression in cabbage was upregulated on 4-fold on day 6 and 6.5-fold in day 12 and 2-fold change on day 6 and day 12 at lower temperature (Figure 4F). *EIN3* has comparatively lower effect on broccoli and kale as the transcript fold change was lower at room temperature (Figure 4F). Taken together, these data indicate that transcripts involved in ethylene biosynthesis, receptor and signaling show variation among these *Brassica* vegetables under two different conditions indicating that they respond variably to the storage treatments.

## 3. Discussion

The study was performed to investigate the effect of storage temperature treatments within the *Brassicaceae* species including cabbage, broccoli and kale. We discovered that senescence progression and postharvest shelf-life is determined by combination of SAGs, ethylene and environmental conditions. Our results showed that temperature is not the only limiting factor in delaying senescence as the postharvest shelf life was different within the brassicas when subjected to the same storage temperature. In *Brassicaceae* species such as cabbage, broccoli and kale, the senescence progression is regulated at different levels of genetic and physiological environment. Broccoli and kale have relatively higher respiration rates which enables the master utilization of substrates and therefore enhances the respiration process enabling the early yellowing during the postharvest storage at room temperature (Figure 1). Our results from expression profiling studies demonstrated that transcript patterns changed in one of the SAGs where *ORE15* was much higher in kale at room temperature in comparison to cabbage and broccoli. *ORE15*, is one of the potential transcription factors associated with increasing senescence and was reported as a negative regulator. There are increasing prospects of higher productivity using *ORE15* for crop improvement strategies as it can further help to extend the postharvest storage of leafy vegetables [14]. In addition, certain phytohormones such as ethylene is primarily responsible for promoting senescence, although several other hormones have also been reported for their role in senescence [27]. However, alternative application is the 1-methylcyclopropene, an effective inhibitor of ethylene action and mimics its activity in order to alleviate the postharvest problems like early senescence. For example, 1-MCP treatment in pak-choy (lower shelf-life like kale and broccoli) delayed the senescence and yellow of leaves by maintaining the chloroplast integrity during the storage at 20 °C [28]. Additionally, high throughput studies have identified hundreds of potential SAGs, some of which have been further demonstrated to have roles in senescence using functional genomics and reverse genetics approaches [15].

In summary, we explored the functional divergence of SAGs and ethylene-induced senescence genes during the senescence mechanism in postharvest cabbage, broccoli and kale species. A heat map of the transcript levels of the SAGs and ethylene-related genes shows the gene expression patterns summarized in Figure 3D and Figure 4G,H. We found that the expression of ethylene-independent SAGs, *ORE15*, *SAG12* and *NAC29*, were rapidly increased at higher levels during room temperature than under cold storage in all the three vegetables. However, the expression patterns of those SAGs varied at cold treatment (Figure 3D). These results suggested that divergent transcript expression of SAGs in *Brassica* vegetables might lead to the differential senescence mechanism that progresses through the interactions between senescence-related factors and environmental factors. Further, by mining the expression pattern of ethylene biosynthetic, metabolic and signaling genes, we gained insight into the species and tissue-specific variations observed among cabbage, broccoli and kale under two temperature conditions (Figure 4G,H). This study has enhanced our current knowledge that different species even within the *Brassicaceae* family respond to senescence in various ways.

## 4. Materials and Methods

### 4.1. Collection of Plant Materials

Green cabbage heads, broccoli florets and kale were purchased from Publix (Gainesville, FL, USA) and selected considering their uniform size, shape and color. These vegetables were harvested at the same time from a local farm and considered to be at the same physiological age. For the cabbage heads, the exterior leaves (with a dark green apex) were detached with a sharp scalpel, washed with running water and gently dried. Postharvest senescence was studied during storage in the dark at two temperatures: cold (4 °C, cold storage) and room temperature (RT, 25 °C, 70–80 % relative humidity in a plant growth chamber). Three biological replicates from each material (leaves from cabbage and kale; floret tissue from broccoli) were used for each sampling day. Tissue samples were collected from the cabbage leaves at days 1, 3, 6, 9, and 12 and wrapped in aluminum foil. Samples were immediately stored in liquid nitrogen followed by long-term storage at −80 °C.

### 4.2. Measurement of Moisture Content

Fresh weight was monitored by measuring the weight of the cabbage leaves, broccoli heads and kale leaves using three biological replicates from each time point for each storage condition. Five time points were chosen, viz. day 1, 3, 6, 9, 12. It was calculated by averaging the fresh weight changes for the biological replicates from each time point. Significant differences were calculated using student t-test between the cabbage leaves stored at 25 °C and 4 °C.

### 4.3. Measurement of Relative Water Content in Cabbage, Broccoli and Kale

Relative water content (RWC) was measured in cabbage, broccoli and kale with four biological replicates on every other day in nine days. The method has been adapted from Bars and Weatherly, 1962 [26]. During each time point, fresh weight was measured from all replicates of the *Brassica* vegetables and subsequently placed in oven for drying at 75 °C for 24 h to measure the dry weight of the samples. Further, the samples were hydrated in fresh water for 3 h to measure the turgid weight. All weights are performed in milligrams calculations: given below is the formula being used to analyze RWC: RWC (%) = [(FW − DW)/(TW − DW)] × 100; (where FW: Sample fresh weight, TW: Sample turgid weight, DW: Sample dry weight)

### 4.4. RNA Extraction and cDNA Synthesis

The entire cabbage head, broccoli head and kale leaves were stored at room temperature or 4 °C to evaluate the postharvest senescence mechanism. Four biological replicates for each storage temperature were kept for a period of 12 days. Three time points, essentially days 1, 6, and 12, were chosen for expression studies based on major morphological changes. Total RNA was extracted using Trizol (Ambion Life Technologies, Grand Island, NY, USA) from three independent biological samples for each temperature. After DNase treatment, RNA was quantified using Nanodrop (Applied Biosystems, Foster City, CA, USA). RNA (1 µg) was used for complementary DNA synthesis (Bio-Rad, Hercules, CA, USA). cDNA was further diluted to 1:5 and 2 µL was used as template for the real time PCR reaction.

### 4.5. Gene Expression Studies by RT-qPCR

Expression studies were performed on samples collected from cabbage, broccoli florets and kale leaves after 1, 6, and 12 days of storage. Relative transcript expression was investigated using three technical replicates from each of the biological replicates from the three time points on day 1, 6 and 12. Genes for ethylene biosynthesis, *ACO* and *ACS*, signaling genes *EIN2*, *EIN3* and perception genes like *CTR1* and *ETR1*, and the SAGs, *ORE15, NAC29* and *SAG12* were analyzed using the primer sequences listed in Table 1. *Actin2* was used as an internal reference control (Table 1). Each reaction was 10 µL using SYBR green reagents from Power up SYBR green kit (Thermo-Fisher, Grand Island, NY, USA). Each gene was amplified in three separate reactions using a Thermofisher (Applied biosystems, Foster City, CA USA) thermocycler with the following conditions: 95 °C for 10 min, 45 cycles for 95 °C for 30 s, and 60 °C for 30 s. Relative gene expression was calculated by the ΔCT method. Quantitative PCR experiments were repeated for three times.

### 4.6. Statistical Analysis

Statistical Analysis of results was performed using Microsoft Excel for windows. Data was analyzed for hypothesis testing, using one-tail, paired sample Student’s *t*-test, followed by the comparison of means checked for least significant difference rejecting null hypothesis for *p*-values < 0.05 (statistically significant). The *t*-tests were calculated for any significant differences between day 1 and day 6 and day 1 and 12.

## 5. Conclusions

Recent advancements in horticultural research have greatly enhanced our knowledge about how senescence is affected by developmental and environmental processes. The study of gene expressions has illustrated that ethylene-related genes and SAGs played diverse roles in senescence. Ethylene receptors and downstream signals generate different responses in different species, so detailed studies must be done in each important agricultural crop. It remains yet unclear what key regulatory factors functions during postharvest senescence. Additionally, variable rates of moisture retention increases the substrate utilization and hence enhancing the senescence process. This expression study revealed how some internal factors related to ethylene signaling and external factors like storage temperature can affect the shelf life in *Brassicaceae* species and would further help in maintaining their commercial quality. It is also concluded that expression of SAGs was strongly linked with storage at room temperature however variation in transcript levels of genes associated with ethylene signaling could affect the degree of senescence in brassicas. Finally, from the overall results, it is inferred that cabbage has a higher shelf life than broccoli followed by kale which indicates that senescence events occurred faster in kale among these species.

## Figures and Tables

**Figure 1 ijms-22-00839-f001:**
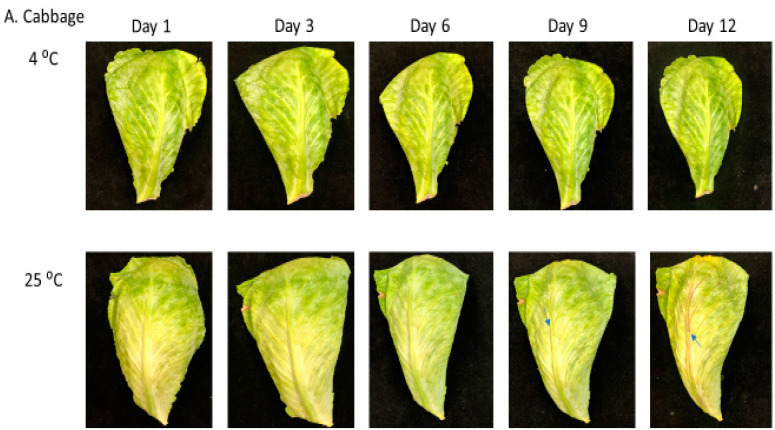
Images of detached green cabbage (**A**) broccoli heads (**B**) and kale leaves (**C**) during storage at room temperature (25 °C) and under refrigeration (4 °C) for over twelve days. Green cabbage leaves displayed browning at day 6 at room temperature and on day 9 under cold storage, whereas broccoli and kale started showing visible changes on day 3 at 25 °C. Obvious yellow spots in broccoli heads can be observed on day 6 at room temperature.

**Figure 2 ijms-22-00839-f002:**
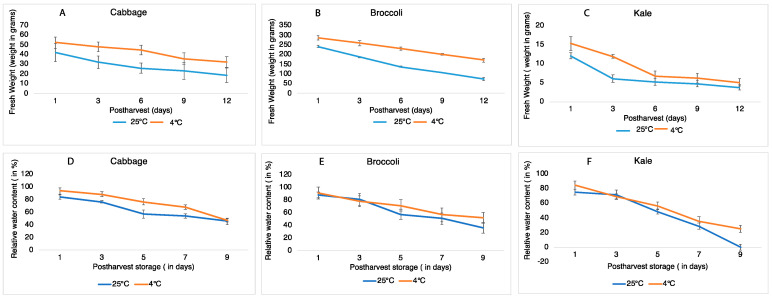
Fresh weight and relative water content in cabbage leaves, broccoli florets and kale leaves stored at 25 °C (blue) and 4 °C (red) in 12 days. (**A**–**C**) showing fresh weight (in %); (**D**–**F**) relative water content in three vegetables. Data represents means ± SE bars (n = 4) for each day).

**Figure 3 ijms-22-00839-f003:**
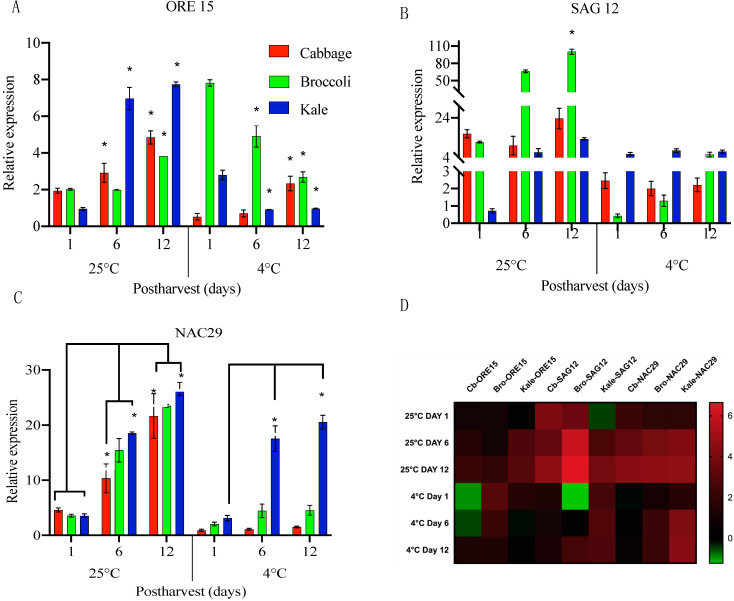
Relative expression of senescence-associated genes *ORE15* (**A**), *SAG12* (**B**), and *NAC29* (**C**) in cabbage leaves (red), broccoli (green), and kale (blue) stored at 25 °C and 4 °C for 12 days. (**D**) Effect of storage temperature (25 °C and 4 °C) on selected SAGs represented as heap maps. Fold change is expressed as log 2 ratio of transcript levels represented in the form of colors. Red color indicates upregulation and green color indicates downregulation. The heat map was constructed using graph-pad prism 8.4 software with the normalized expression profiles. Data represent means ± SE bars (*n* = 3 leaves for each day). Asterisks (*) if any indicate statistically significant differences from day 1 (control) to treated samples (day 6 and day 12) for each time point (*p* < 0.05).

**Figure 4 ijms-22-00839-f004:**
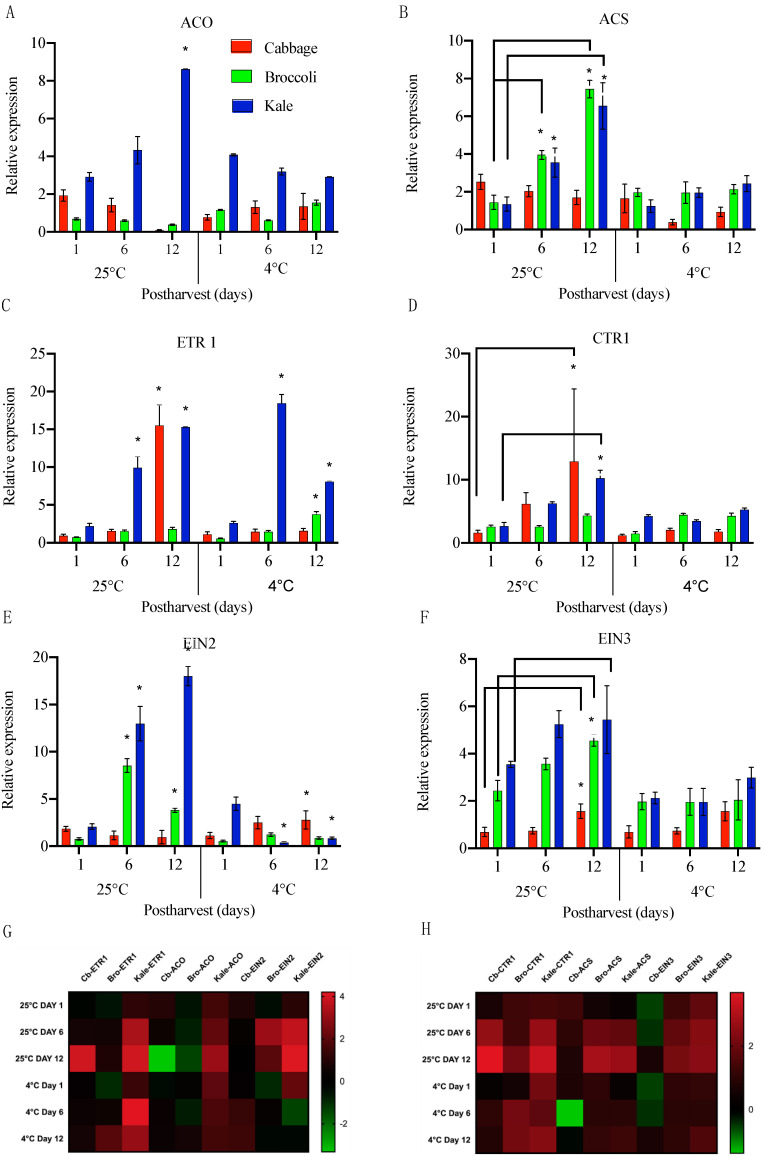
Relative transcript levels of ethylene biosynthesis ((**A**)-*ACO* and (**B**)-*ACS*), perception ((**C**)-*ETR1* and (**D**)-*CTR1*), and signaling ((**E**)-*EIN2* and (**F**)-*EIN3*) genes in cabbage, broccoli and kale stored at 25 °C and 4 °C on day 1, day 6 and day 12. Data represents means ± SE bars (n = 3 for each time points). Asterisks (*) if any indicate statistically significant differences between day 1 (control) and either day 6 or day 12 under the same storage condition (*p* < 0.05). (**G**,**H**). Effect of storage temperature (25 °C and 4 °C) on selected ethylene-related genes represented as heap maps. Fold change is expressed as log2 ratio of transcript levels represented in the form of colors. Red color indicates upregulation and green color indicates downregulation. The heat map was constructed using graph-pad prism 8.4 software with the normalized expression profiles.

**Table 1 ijms-22-00839-t001:** List of primers used for real-time PCR analysis of ethylene biosynthesis and signaling genes and selected senescence-associated genes. Primers were designed using sequences from the *Arabidopsis* genome and designed with the help of Integrated DNA technologies (IDT) software.

SAG12-RT-FP	TGGAATTGAAGGAGGTGGTTT
SAG12-RT-RP	GCCGTATCCAATCGCAGTTA
EIN2-RT-FP	GTGCGTCTTATGGTCGGTTA
EIN2-RT-RP	CTCTGTAGCCTCCTCTTGATTG
ACTIN2-RT-FP	CTTGCACCAAGCAGCATGAA
ACTIN2-RT-RP	CCGATCCAGACACTGTACTTCCTT
EIN3-RT-FP	AAGGGAGTGGTGGAAAGATAAG
EIN3-RT-RP	GTCGGTCCAATCGGGTTATT
CTR1-RT-FP	AAATCCGGCTCAGGTTGTAG
CTR1-RT-FP	GGTCCAACAACCCTCGATTAT
ETR1-RT-FP	TGAGTTCAAACGAGGAGTGTC
ETR1-RT-FP	CGGAGAGCGATTTGGTAGTT
ACO_FP_RT	AGATGATGGAAGTGATGGATGAG
ACO_RP_RT	GATGAGGACAAGGAGGGTAATG
ACS_FP_RT	CACGAGACGGTTGCTTTCT
ACS_RP_RT	CCGGTTCTCCATCTCAAATCTC
NAC29_FP	GCAGAGAGAAGAACTGAAGAAGAG
NAC29_RP	AGAGACGGGTCCCATGTAAT
ORE15_FP	CATCGCTCTCATCCTCTTCTTC
ORE15_RP	AGCTTCTCAAGATCACTCAACC

## Data Availability

Data is contained within the article.

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

*thaliana* leaves. Physiol. Plant.

