# Peer review of "Varied Expression of Senescence-Associated and Ethylene-Related Genes during Postharvest Storage of Brassica Vegetables"

_ijms, 2021, doi:10.3390/ijms22020839_

Round 1

Reviewer 1 Report

The revisions are hard to follow I don't see the author's responses to my concerns raised with previous versions.

Author Response

Dear Reviewer1,

      We have perused the reviewers comments and strongly appreciate the feedback provided by them in making this manuscript a deemed fit into your journal. Hence, in this regard, we provided new data to address reviewers` comments  below as well and revised the manuscript with marked changes in the grey color as highlighted in the revised manuscript.

Reviewer 1:

  1. We agree with the reviewers that it is more perfect to characterize moisture content by quantifying the Relative Water Content (RWC). Therefore, we measured RWC to describe water status among Brasscia vegetables at successive time-points. We found the data consistancy between weight loss and RWC. In the results, we observed that relative water content in kale was significantly lower at room temperature and even at cold storage in comparison to broccoli or cabbage suggesting that kale has lower water retention capacity. Further descriptions are highlighted in grey in our revised manuscript.
  2. The reviewer commented on the limitation that only a small number of genes have been tested for transcript studies.. To address the comments, we selected additional SAGs as well as key genes in ethylene biosynthesis, perception, and signaling including NAC29, ACS, CTR1, and We characterized  their expression patterns in postharvest Brassicavegetables and provided transcriptional comparison among  those genes as well.
  3. For the conclusion, we emphasize that this study focussed on the funtional divergence between ethylene-independent SAGs and ethylene metabolic and signaling genes. The results could potentially help us to understand the diverse mechanisms in senescence-associated regulators and ethylene-induced senescence pathway.
  4. We agree with the reviewer stating the fact that there are tissue-type differences between leafy vegetables and inflorescence vegetable. However, this research is focused on understanding the molecular mechanism of postharvest senescence which further extend our knowlege on the shelf-life and to improve our commercial quality and values in Brassica plants. 

We anticipate your kind interest into our revised submission to be considered for your journal. Please feel free to address any correspondence concerning this manuscript to our emails at yogesh.ahlawat@ufl.edu, and tieliu@ufl.edu. Thank you for the time and consideration.

Sincerely,

Tie Liu, PhD

Assistant professor,

Postharvest genomics

Horticultural Sciences, University of Florida, USA

Reviewer 2 Report

The resubmitted paper is much more better prepared, however I advised some minor revision. The Figure 3 and 4 should be organized in some other way, as in the present form they look a bit chaotic. In the description of relative water content I advise to change W stated for fresh weight into FW, as usually used in literature. Such sentence as: “Transcriptomics has identified differentially expressed genes with roles in senescence” should be modified, as analysis of several genes should not be named transcriptomic, but analysis of genes expression. Moreover the identification was done by the authors not by the method of study. Yue can add heat map analysis of all studied genes for each of the species. Such visualization of the results show in more familiar way groups of genes which are regulated in similar manner.

Author Response

Dear Reviewer 2,

Please find enclosed our revised manuscript. We thank the two reviewers for their comments. We have revised the manuscript accordingly and provide specific answers below. 

Please find enclosed our revised manuscript. We thank the two reviewers for their comments. We have revised the manuscript accordingly and provide specific answers below. 

  1. The resubmitted paper is much more better prepared, however I advised some minor revision.

Thank you very much for your comments.

  1. The Figure 3 and 4 should be organized in some other way, as in the present form they look a bit chaotic.

Thank you for your suggestions. We reformatted the figure 3 and 4, reduced labelling and numbering, and added a heat map to summarize the results.

  1. In the description of relative water content I advise to change W stated for fresh weight into FW, as usually used in literature.

Done.

  1. Such sentence as: “Transcriptomics has identified differentially expressed genes with roles in senescence” should be modified, as analysis of several genes should not be named transcriptomic, but analysis of genes expression. Moreover the identification was done by the authors not by the method of study.

Sentence was modified as the reviewer`s suggestion.

We rephrased the sentence, “The study of gene expressions has illustrated that ethylene-related genes and SAGs played various roles in senescence.”

  1. Yue can add heat map analysis of all studied genes for each of the species. Such visualization of the results show in more familiar way groups of genes which are regulated in similar manner.

We added heat maps to Figure 3 and 4 to display gene expression patterns in groups of genes.

We greatly appreciate your feedback.

Sincerely,

Tie

Assistant Professor

Plant Molecular and Cellular Biology Program

Horticultural Sciences Department

Genetics Institute

University of Florida, Gainesville, FL, USA

This manuscript is a resubmission of an earlier submission. The following is a list of the peer review reports and author responses from that submission.

Round 1

Reviewer 1 Report

The review of the manuscript draft entitled “Varied expression of senescence-associated and ethylene-related genes during postharvest storage of Brassica vegetables” submitted to IJMS by no. ijms-991184. The paper is nicely written and consistent, however in my opinion does not suit to this journal. It cover analysis of genes expression only for 5 genes – some from ethylene metabolism. There are also some basic physiological analysis. There is lack of measurement of ethylene production. Also one of the main remark is the fact, that authors compared 2 different organs, as in case of broccoli they use inflorescence for the study, and for kale and cabbage leaves. The fresh mass of the organs, which function also as a water reservoir also differ between particular species. The authors monitored water loss in particular species, but in my opinion it is not enough. I advise to monitor also water potential and relative water content. The RWC in my opinion much more better describes changing in water status within plant organs. In such analysis it is nearly sure that the authors would observed differences between samples. However in journal with such high IF as IJMS some focus on mechanistic hypothesis should be also paid. Even the authors have written in L348 that detailed studies must be done – the present study is not detailed. In my opinion this paper could be transmitted to f.eg. Agronomy MDPI, where it will be more relevant, as the results may also implicate to commercial quality of brassica species. Also such conclusion as stated in L126-127 “this study showed that lower temperature inhibits the expression of ethylene biosynthetic genes, slowing the progression towards senescence” is commonly known in literature in this filed.

Some minor remarks:

There is lack of country name in authors affiliation.

The Latin name Arabidopsis should be written italics, but is some lines is not: f.eg.L60, 62, 72, 89…

In my opinion the enzyme names should be written with lower case vide L68, 85, 86

I advise to add a citation of the paper published in IJMS in 2019 by Ramirez et al. (Int. J. Mol. Sci. 2020, 21(6), 1998; https://doi.org/10.3390/ijms21061998).

The Figures 2, 3, and 4 could be a bit larger.

Reviewer 2 Report

This study is aimed to understand the molecular basis of postharvest senescence in Brassica vegetables. The main evidence provided in this study are water losses of vegetables and expression pattern of 5 genes associated with senescence or ethylene biosynthesis. In first place, this evidence is not sufficient to conclude this study

Major concerns

  1. Temperature or ethylene is regulating the above said genes are not provided?
  2. Fig 2; water loss is more in vegetables stored at 4 degree than room temperature (cabbage, broccoli and to some extent in kale). So authors claim in section 2.1 is wrong?
  3. 3. Line no 132, .......likely due to chlorophyll breakdown..........however no evidence is provided?
  4. Is there any correlation among temperature ,water loss and gene expression?. IN fact, there were biases in gene expression (Fig 3). A: broccoli at day 6 is slightly reduced or remain same as that of day 1; At 4 degree and day 1 expression of ORE is so high on day 1?...samples collected from similar environment/sources?
  5. Line no 189.says........SAG12 was highly reduced at RT.....However fig 3B shows that at day 6 (cabbage) reduced..explain
  6. In fig 4A; at RT, day 12;cabbage-ACO expression reduced....meaning ethylene production reduced at RT?
  7. In fig 4 BETR1 expression in Broccoli is not altered in either day 6 or 12..Similar discrepencies  also found in 4C
  8. Fig 5 is repetitive.delete it.
  9. There were also grammetrical and typographical errors found in the manuscript.